# Directed evolution of a TNA polymerase identifies independent paths to fidelity and catalysis

Mohammad Hajjar [1,5], Victoria A. Maola[1,5], Joy J. Lee [1,5], Manuel J. Holguin [1], Riley N. Quijano[1], Kalvin K. Nguyen[1], Katherine L. Ho [1], Jenny V. Medina[1], Elionel Botello-Cornejo [1], Bhawna Barpuzary [1], Nicholas Chim [1] ✉ & John C. Chaput [1,2,3,4] ✉

Directed evolution facilitates functional adaptations through stepwise changes in sequence that alter protein structure. While most campaigns yield solutions that maintain the framework of a rigid protein architecture, a few have produced enzymes with more notable structural differences. One example is a polymerase that was evolved to synthesize threose nucleic acid (TNA) with near-natural activity. Understanding how this enzyme arose provides a model for studying pathways that guide enzymes toward more productive regions of the fitness landscape. Here, we trace the evolutionary trajectory of an unnatural polymerase by solving crystal structures of key intermediates along the pathway and evaluating their biochemical activity. Contrary to the view that fidelity is a product of increased catalytic efficiency, we find that accuracy and catalysis are decoupled activities guided by separate ground-state and transition-state discrimination events. Together, these results offer a glimpse into the forces responsible for shaping the emergence of new enzyme functions.

Directed evolution is a powerful approach for generating enzymes that are needed to drive future applications in biotechnology and medicine[1–3]. Through recursive cycles of diversification and selection, populations of molecules adapt to novel biochemical functions that lie outside the sphere of activities found in nature[4]. Though effective, most protein evolution studies are unable to match the catalytic efficiency of natural enzymes[5]. The central challenge lies in understanding how to reshape the active site pocket to accommodate noncognate substrates while preserving the catalytic activity of the native enzyme fold. Although aspects of this problem have been studied by examining the products of directed evolution campaigns[6–9], understanding the molecular basis of functional adaptation requires structural information about key intermediates along the pathway, each

representing a critical step toward the enzyme's final, functional form[10]. By studying the evolutionary journey from a low-activity variant to a highly functional enzyme, it may be possible to learn basic design principles that could accelerate future selections or expand the predictive power of AI algorithms to include dynamic enzymes with complicated reaction mechanisms.

In recent work, we demonstrated the directed evolution of a polymerase, called 10–92, that can synthesize α-L-threofuranosyl nucleic acid (TNA) with near-natural activity (Fig. 1)[11]. Biochemical studies reveal that 10–92 achieves a catalytic rate of -1 nt s⁻¹ and >99% fidelity by sequentially adding TNA triphosphates (tNTPs) to the 2′-hydroxyl group of the growing primer, yielding an unnatural 3′ → 2′ linked TNA product (Fig. 1a). The enzyme was discovered

[1]Department of Pharmaceutical Sciences, University of California, Irvine, CA, USA. [2]Department of Chemistry, University of California, Irvine, CA, USA. [3]Department of Molecular Biology and Biochemistry, University of California, Irvine, CA, USA. [4]Department of Chemical and Biomolecular Engineering, University of California, Irvine, CA, USA. [5]These authors contributed equally: Mohammad Hajjar, Victoria A. Maola, Joy J. Lee. ✉e-mail: nchim@uci.edu; jchaput@uci.edu

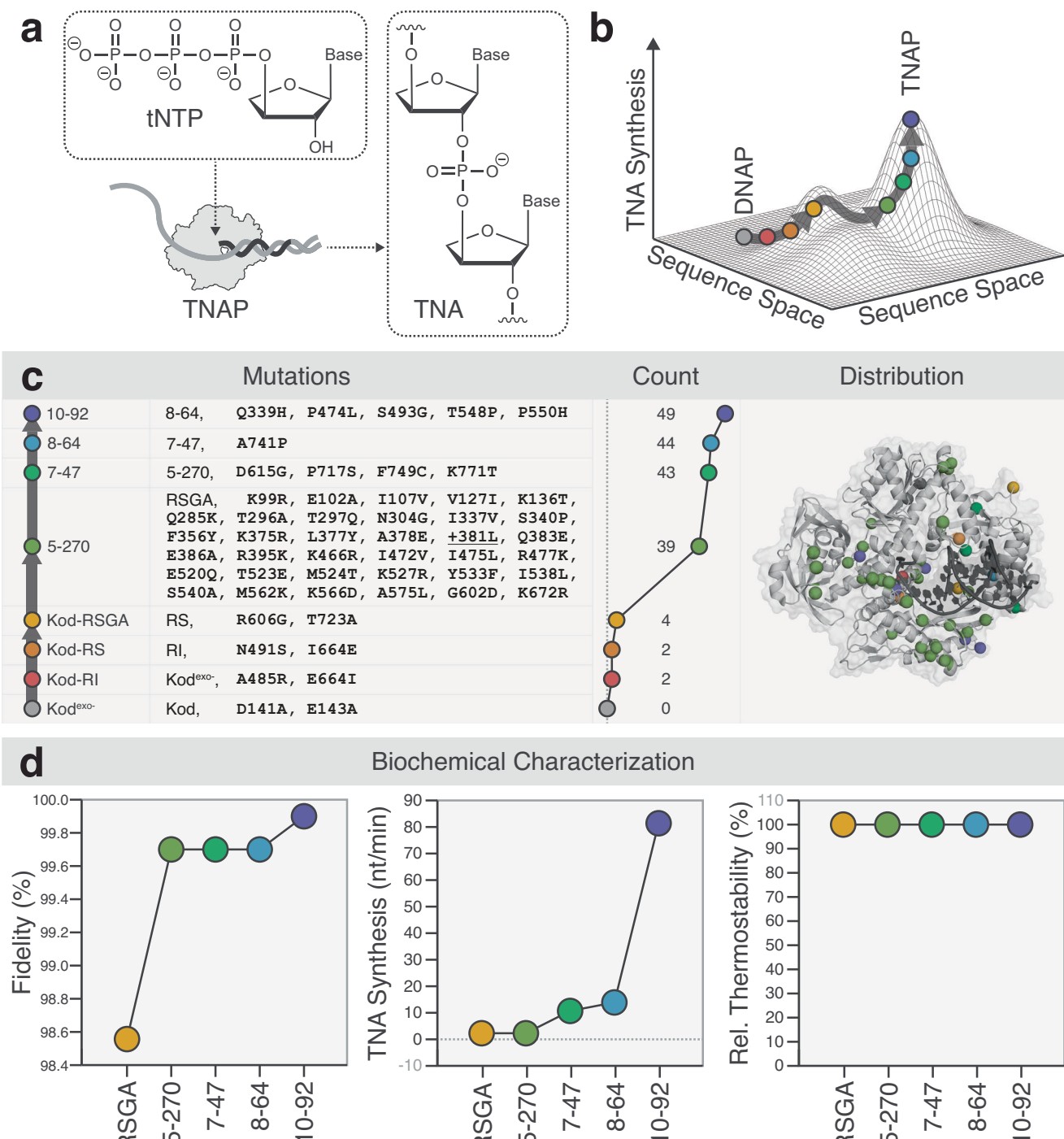

**Fig. 1 | Evolution of the 10–92 TNA polymerase. a** Schematic representation of a TNA polymerase (TNAP) copying DNA into TNA by adding tNTPs to a growing primer. **b** A hypothetical fitness landscape conceptualizing the path taken from a family of natural DNAPs to a highly evolved TNAP. **c** TNAP generations are color-coded relative to Kod DNAP. Mutations are counted and spatially mapped onto the structure of Kod DNAP (PDB: 5OMF) with the DNA duplex shown in black. **d** Biochemical analysis of successive generations of TNAPs from Kod-RSGA to 10–92. Quantitative scatter plots display the values observed for catalysis, fidelity, and thermal stability. Source data are provided in Supplementary Fig. 6 (fidelity) and additional Source data files (kinetics and thermostability).

using the massively parallel screening technique of droplet-based optical polymerase sorting (DrOPS)[12,13] to query a large combinatorial library prepared by recombining the sequences of four hyperthermophilic archaeal B-family DNA polymerases (DNAP) (*Thermococcus kodakarensis*, Kod; *Thermococcus gorgonarius*, Tgo; Pyrococcus Deep Vent, DV; and Thermococcus 9°N, 9°N) (Fig. 1b, c). Each polymerase carried the exonuclease silencing mutations (D141A and D143A) and four beneficial mutations (A485R, N491S,

R606G, and T723A) found in Kod-RSGA, our previous best TNA polymerase (TNAP)[14].

A crystal structure of the 10–92 TNAP uncovered large movements in the protein architecture that improved substrate binding, including positional rearrangements of α-helical secondary structures (up to 7 Å) that bring the thumb subdomain closer to the paired region of the DNA duplex and other large movements that realign the catalytic triad in the palm subdomain (Supplementary Figs. 1 and 2)[11].

Together, these structural changes enabled the transition from an enzyme capable of highly efficient DNA synthesis to one that is now specialized for TNA synthesis (Supplementary Fig. 3). Whether these structural changes arose early or late in the evolutionary process remains to be established, as do the underlying mechanisms responsible for achieving enhanced catalytic activity.

Here, we study the evolutionary trajectory of the 10−92 TNAP by solving crystal structures of key intermediates along the pathway and evaluating their biochemical activity. Crystal structures were solved as binary and closed ternary complexes, and biochemical data were collected for fidelity, catalytic efficiency, thermal stability, and substrate specificity. Among the more notable trends is the observation that fidelity and catalysis appear to occur as decoupled activities, possibly guided by separate ground- and transition-state discrimination events. Molecular insights into these processes were obtained from crystal structures that were used to map the process of active site refinement across successive generations of the enzyme. Together, these findings provide a detailed mechanistic framework for understanding how directed evolution reshapes enzyme function to achieve accurate and efficient catalysis.

## Results
### Biochemical analysis of the evolutionary intermediates
To follow the biochemical changes that occurred along the evolutionary path, we evaluated the polymerases that most influenced the evolution of the 10−92 TNAP for fidelity, activity, thermal stability, and substrate specificity. This analysis included Kod-RSGA as well as intermediates 5−270, 7−47, and 8−64 (Fig. 1c and Supplementary Fig. 4), the most active variants identified in rounds 5, 7, and 8, respectively, and the parent sequences used for further diversification[11].

Our analysis of fidelity and catalysis reveals a general trend in which steep gains in fidelity occur in earlier generations of the enzyme, whereas analogous improvements in activity are delayed until later stages of the evolutionary trajectory (Fig. 1d; Supplementary Figs. 5−7 and Supplementary Table 1). However, it should be noted that we do observe a concomitant improvement in fidelity as catalysis improves, presumably due to fine-tuning of the active site near the end of the evolutionary trajectory. Nevertheless, the overall trend of early improvements in fidelity and later improvements in catalysis caught our attention as it appears to counter the prevailing view that a selection for improved catalytic activity through the use of progressively shorter incubation times, as was the case for 10−92[11], should indirectly enrich for enzymes with higher fidelity[15,16]. This is because less faithful polymerases are more likely to stall at misincorporation sites, thereby failing to complete the extension within the allotted time[17–22]. Thus, variants that maintain rapid synthesis under stringent incubation times are expected to exhibit the combined properties of enhanced catalytic efficiency with increased fidelity. While this interpretation is consistent with the selection results overall, our biochemical analysis of the evolutionary intermediates indicates that variants can emerge that exhibit enhanced fidelity without a proportional gain in catalytic activity. This observation implies that fidelity and catalysis may have determinants that can be separately tuned during the early stages of evolution.

We also found that all of the polymerases tested retain their extreme thermostability, showing no detectable loss in TNA synthesis activity after prolonged exposure to 90 °C for 6 h (Fig. 1d and Supplementary Fig. 8). This result highlights the exceptional structural integrity of the variants and challenges the view that directed evolution inevitably compromises protein folding stability[5]. Such trade-offs are particularly relevant in enzyme engineering campaigns, where enhanced activity often comes at the expense of reduced thermostability[23]. Remarkably, the 10−92 TNAP, with its 51 amino acid mutations relative to its closest natural homolog, Kod DNAP (Fig. 1c),

exhibits no measurable loss of thermostability (Supplementary Fig. 8). This observation supports the view that protein stability promotes evolvability[24], which may be a result of the thermal lysis step of the DrOPS selection protocol[25].

Finally, the 10−92 TNAP also exhibits a striking shift in substrate specificity, defined here as the ability for the polymerase to distinguish TNA versus DNA substrates based on their catalytic rates of incorporation. Kinetic analysis across generations of the polymerase reveals a consistent reluctance of the enzymes to relinquish their ancestral DNA synthesis activity until the final stages of evolution, at which point an inversion (13-fold) in substrate specificity occurs, favoring TNA synthesis over DNA synthesis (Supplementary Fig. 7). This inversion of substrate preference marks the transition of the enzyme from a broadly acting generalist to one that is increasingly specialized for TNA synthesis on DNA templates. Such a shift is a hallmark of functional divergence, arising from the accumulation of mutations that reconfigure the active site toward a new function.

### A binary complex capturing an unexpected preorganized state
As an initial step toward solving the puzzle of how 10−92 emerged as a highly active enzyme from a diverse pool of synthetic homologs, crystal structures of major intermediates identified along the evolutionary pathway were first solved as binary structures of the post-catalytic complex with a bound DNA primer-template (P/T) duplex that was enzymatically extended with two TNA residues (3′-tTtA-2′) at the 3′ terminus. Four structures (5−270, 7−47, 8−64, and 10−92) spanning a resolution limit of 2.17−2.56 Å were determined by molecular replacement (Supplementary Table 2). Consistent with all known archaeal B-family DNAPs, the enzymes adopt a disk-shaped architecture (Supplementary Figs. 9 and 10) defined by an N-terminal domain (NTD), an exonuclease domain, and a catalytic domain comprised of the finger, palm, and thumb subdomains[26]. In all cases, the duplex is well resolved and tightly bound between the palm and thumb subdomains (Supplementary Fig. 9).

Surprisingly, comparative structural analysis reveals an unexpected, closed conformation for the finger subdomain in all four of the TNAP structures. In this conformation, helices α14 and α15 are shifted by ~22° relative to their position in the open binary structure of wild-type Kod DNAP (Fig. 2a−e)[27]. This finding contrasts with all known binary structures of replicative DNAPs, which typically exhibit an open finger conformation ready to accept the incoming triphosphate[28]. Until now, the closed state has only been observed in structures of ternary complexes trapped in a catalytically competent state of the reaction cycle (Fig. 2b, c)[29,30].

While it is possible that one could consider this observation a crystallization artifact, several lines of evidence argue against this option. First, the TNAP variants (5−270, 7−47, 8−64, and 10−92) crystallized under two distinct precipitation conditions, suggesting that the conformation reflects a bona fide structural feature rather than a condition-specific artifact. Second, the closed binary conformation is absent in previous binary structures of Kod-RI[31] and Kod-RSGA[32] (Supplementary Fig. 11), further strengthening the view that the conformational difference arises from sequence changes rather than crystallization chemistry. Finally, the closed binary conformation emerged in variants that also exhibit changes in their biochemical activity, providing a functional correlation that supports its structural relevance.

To understand the molecular basis of this distinct conformation, we analyzed the interdomain interactions stabilizing the closed state. In the binary form of natural Kod DNAP, only limited contacts exist between the exonuclease domain and finger subdomain (Fig. 2f)[27], whereas noticeably more contacts are observed as Kod transitions to a closed ternary conformation, supporting a catalytically competent state with the dNTP poised for chemical bond formation (Fig. 2g)[26]. Contrasting the natural system, the closed binary conformation of the TNAP intermediate observed in 5−270, 7−47, 8−64, and 10−92, exhibits

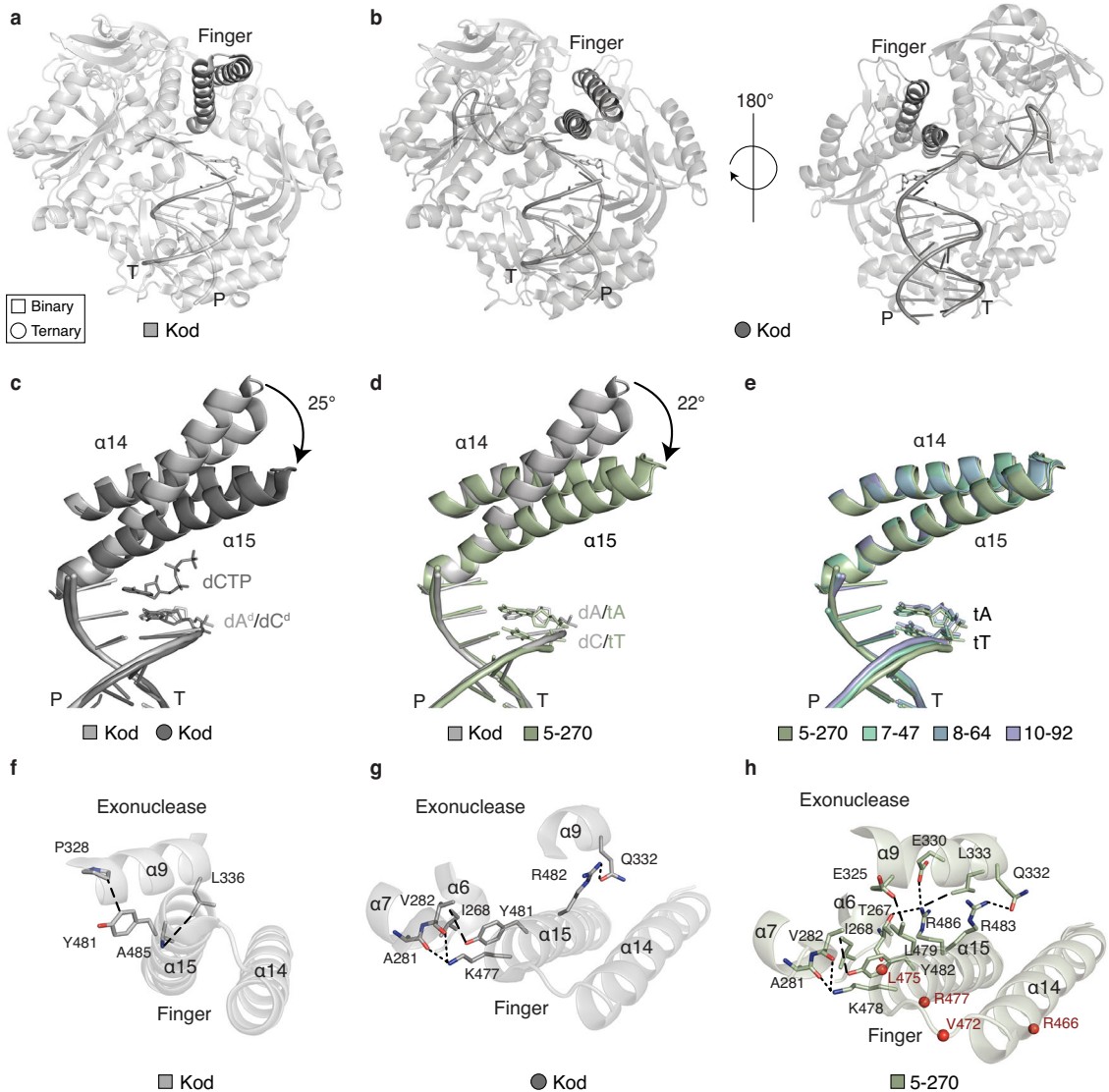

**Fig. 2 | The finger subdomain of TNA polymerases. a, b** Global architecture of Kod DNAP captured in an open binary complex with dsDNA (A, PDB: 4K8Z) and closed ternary structure in complex with dsDNA and dATP (B, PDB: 7OMB). The finger subdomain is shown in black. **c** Overlay of the open (light gray) and closed (dark gray) conformations of the finger subdomain found in Kod DNAP. **d** Overlay of the finger subdomain of Kod DNAP with 5–270, both crystallized as a binary complex. **e** Overlay of the finger subdomains of 5–270, 7–47, 8–64, and 10–92, each

crystallized as a binary complex. Molecular interactions observed in the finger subdomain of Kod binary (**f**), Kod ternary closed conformation (**g**), and 5–270 binary (**h**). Residue labels: natural residues found in Kod DNAP (black); acquired mutations (red spheres, labels). Short dashed lines denote electrostatic interactions, long dashed lines show hydrophobic interactions. TNAP generations are color-matched to Fig. 1.

a robust network of electrostatic and hydrophobic interactions that preorganize the enzyme in a catalytically active conformation despite the absence of a bound tNTP (Fig. 2h).

The structural similarity between the binary TNAP complexes and the closed ternary structure of 10–92 previously solved by X-ray crystallography is striking, particularly in the geometry of the active site, which differs only by the absence of substrate-specific contacts (Supplementary Fig. 12)[11]. Because the closed binary motif first appears in the 5–270 intermediate and is absent in the binary structure of Kod·RSGA[32] (Supplementary Fig. 11), we infer that this conformational switch arose from recombination events that facilitated TNA synthesis by stabilizing the enzyme in a catalytically productive state. In this scenario, the finger subdomain exists in a dynamic equilibrium between the open and closed conformations, reminiscent of natural DNAPs[20]. However, upon correct substrate binding and the ensuing conformational change[19], the pre-organized state would provide a selective advantage by biasing the

enzyme toward the closed conformation, thus offering more time for catalysis to occur prior to substrate release.

## The catalytically active state of the enzyme

To visualize how catalytic activity evolved within the TNAP lineage, we determined the crystal structures of key intermediates in a catalytically competent state with a closed finger conformation and a P/T duplex and tATP substrate bound in the enzyme active site. A key aspect of this effort was the chemical synthesis of a chain-terminating 2′-deoxy-α-L-threofuranosyl thymidine-3′-triphosphate (dtTTP) analog, prepared in 8 synthetic steps from a 3′-protected α-L-threofuranosyl thymidine (tT) precursor using a Barton deoxygenation strategy to remove the 2′ hydroxyl group (Supplementary Figs. 13–16)[33,34]. This substrate enabled us to capture the reaction state immediately following tNTP binding, but before chemical bond formation. Using this approach, we successfully obtained high-resolution crystal structures

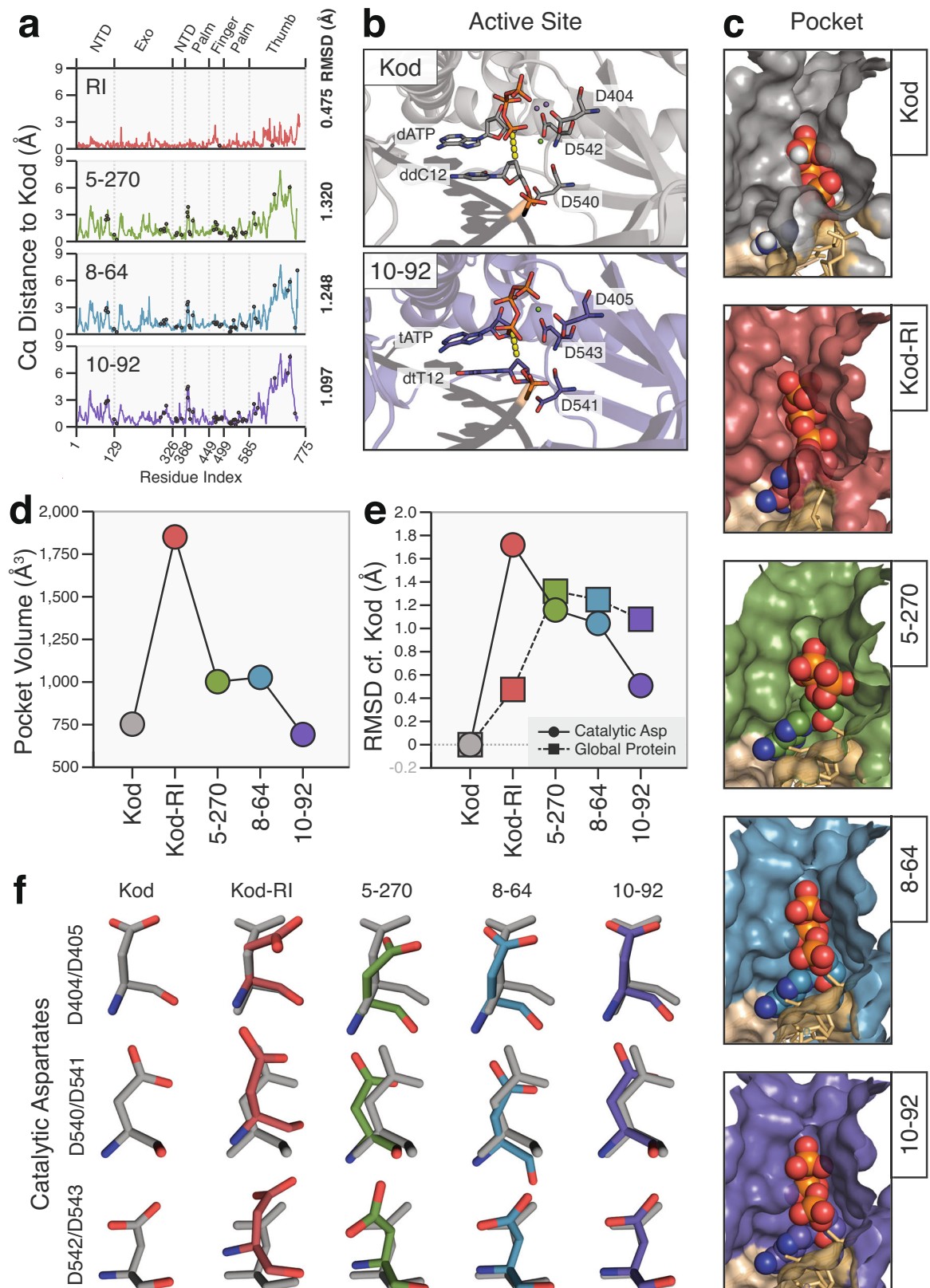

for 5–270 and 8–84, resolving to 3.03 and 2.38 Å, respectively (Supplementary Table 3).

By combining this data with previously solved structures for Kod-RI[31] and 10–92[11], the least and most optimized versions of the enzyme, respectively, a structural framework was established to study the biochemical changes that occurred along the evolutionary path.

Superposition of the four TNAP structures (Kod-RI, 5–270, 8–64, and 10–92) reveals that a global divergence in structure occurred following recombination (Fig. 3a, e and Supplementary Fig. 17). The backbone Cα root mean square deviation (RMSD) relative to wild-type Kod DNAP increases from ~0.5 Å for Kod-RI to ≥1.0 Å for 5–270, 8–64, and 10–92, with movements of secondary structural elements observed in the

**Fig. 3 | Stepwise changes in global and local structural conformation. a** The distance in angstroms from all resolved C-α positions in the ternary structures of TNAP generations relative to their position in Kod DNAP. Black dots indicate the position of the acquired mutations. **b** Catalytic aspartate triads observed in the active site of Kod (top) and 10–92 (bottom) ternary structures. Dashed yellow line indicates the distance from the primer to the alpha phosphate of the incoming nucleotide. Spheres indicate the position of divalent metal ions. **c** Shape-filling representation of the active site pocket of progressive generations of TNAPs. Color code: primer (yellow), template (wheat), and triphosphates are colored by element. **d** Comparison the active site pocket volume (Å³) for Kod, Kod-RI, 5–270, 8–64, and 10–92. **e** Root mean square deviation (RMSD) of C-α positions calculated for global (squares) and local (circles) regions of TNAP generations relative to Kod DNAP. Local RMSD values reflect all-atom calculations for the catalytic triad as shown in (**b**). **f** Stepwise optimization of the catalytic aspartate triad across TNAP generations. Overlays show the residue position in the TNAP relative to Kod (gray) via global structural alignment. TNAP generations are color-matched to Fig. 1.

thumb, palm, and NTDs. These changes correspond to a progressive remodeling of the active site pocket (Fig. 3b–f), which aligns with observed biochemical gains in TNA synthesis activity (Fig. 1d). Surprisingly, most of the 51 acquired mutations found in 10–92 are located >20 Å from the catalytic core (Fig. 3a and Supplementary Fig. 18), emphasizing the importance of long-range structural interactions in fine-tuning the active site geometry[35].

Importantly, deep learning models failed to predict these transitions. Structure predictions by AlphaFold3 yielded only canonical B-family DNAP structures, with open binary conformations and no evidence of the experimentally observed rearrangements (Supplementary Figs. 19–23). This observation underscores a key limitation of current AI-based modeling, which is the inability to accurately capture long-range mutations and evolution-driven conformational shifts that underlie new protein functions. Future improvements in predictive accuracy will require experimental benchmarks and better mechanistic understanding of conformational plasticity during enzyme evolution.

### Reshaping the active site pocket

Our previous crystal structure of the 10–92 TNAP trapped in a closed ternary complex revealed how recombination and molecular evolution enabled substantial active site remodeling to support efficient TNA synthesis[11]. The selected mutations produced local backbone shifts, including high-energy rearrangements, that restructured the active site pocket to promote optimal hydrogen bonding, electrostatic stabilization, and van der Waals complementarity with the bound tNTP. In this conformation, the catalytic aspartate triad (D405, D541, and D543) is precisely arranged to coordinate metal ions and facilitate robust nucleotide transfer chemistry.

By integrating structural data from intermediates along the evolutionary path, we now provide a high-resolution, atomic-level view of how these changes unfolded. A central theme to emerge is the stepwise refinement of the active site geometry, beginning with a disruptive expansion of the active site volume followed by precise, substrate-specific remodeling. For example, the active site volume increases ~250% from wild-type Kod DNAP to Kod-RI, before contracting through 5–270 and 8–64 to a more compact volume of 692 Å³ observed in 10–92, which is approximately 10% smaller than the starting enzyme (Fig. 3c, d and Supplementary Fig. 24). This process of active site refinement allows the enzyme to accommodate the smaller size of the tNTP substrate, noting that TNA derives from a four-carbon threose sugar, as compared to the five-carbon sugar found in DNA[36].

At the local level, analysis of the catalytic triad reveals a clear stepwise refinement across the lineage. Overlaying the active site from successive TNAP generations shows that the catalytic residues undergo a series of positional and rotational movements, supported by Polder maps, that ultimately align the aspartate residues into a catalytic configuration closely mimicking the arrangement found in wild-type Kod DNAP (Fig. 3e, f; Supplementary Figs. 25 and 26). These refinements enable TNA substrate recognition while preserving the structural features essential for metal ion coordination and catalysis.

Our structural analysis reveals that base pair recognition and reaction center geometry emerged as two distinct evolutionary pressures shaping the evolutionary trajectory. The former is reflected in the conformation of the nascent Watson–Crick base pair formed between the incoming tATP and the templating thymine base. Following the active site distortion observed in Kod-RI, base pair geometry was largely restored in 5–270, as measured for the parameters of opening, propeller, and buckle (Fig. 4 and Supplementary Table 4). By comparison, the geometry of the reaction center improved more gradually across generations of the enzyme, with the distance from the C2′ group of the primer to the α-phosphate of the bound tATP steadily decreasing from 5.8–6.7 Å in 5–270 (Supplementary Fig. 27) to eventually 3.9 Å in 10–92. This final value closely approximates the ideal distance of 3.7 Å observed in wild-type Kod DNAP (Fig. 4b). Together, these findings underscore the evolutionary mechanism underpinning TNAP optimization; namely, early gains in fidelity correspond to improved base pair recognition, while later gains in catalytic activity emerge from a fine-tuning of the active site geometry. This structural and functional convergence culminates in a highly specialized enzyme whose performance reflects both macro-scale architectural adaptation and micro-scale atomic-level precision.

### Mutational landscape underlying the activity of 10–92

The mutational landscape underlying the evolution of the 10–92 TNAPs reveals a clear shift from broadly distributed, largely conserved substitutions found in 5–270 to a set of later-arising, non-conserved mutations that cluster in the thumb subdomain and strongly impact TNA synthesis activity (Fig. 5 and Supplementary Table 5). This distribution is unsurprising as the mutations found in 5–270 are the result of homologous recombination, while those in 10–92 derive from focused mutagenesis[11]. Structural analyses show that mutations occur across both surface exposed and buried positions, with most mutations residing far from the catalytic center (Fig. 5), again underscoring the importance of long-range interactions in shaping new protein functions[35]. Computational free-energy predictions indicate that while early substitutions are generally neutral with respect to stability, most of the function-driving mutations obtained from directed evolution are destabilizing (Fig. 5), highlighting the trade-off between structural integrity and catalytic enhancement[23]. Limited reversion analysis of later-stage mutations demonstrates that at least one function-driving mutation is present in each of the key intermediates (Fig. 5)[11]. Together, these data suggest that polymerase specialization for TNA synthesis emerged through a process involving the accumulation of large numbers of mostly neutral mutations via homologous recombination that change the protein architecture, followed by a smaller set of adaptive mutations that refine catalytic activity at the cost of local stability.

Identifying the precise set of mutations responsible for improved fidelity is challenging because 5–270 carries 35 mutations, all derived from natural diversity. These substitutions collectively remodel the enzyme architecture, produce major structural alterations that reshape the active site, improve the geometry of the nascent Watson–Crick base pair of the incoming tNTP, and facilitate formation of the preorganized conformation observed in the binary structures. While this extensive remodeling likely underlies the fidelity gains observed in 5–270, the large number of changes makes it difficult to determine which mutations are adaptive drivers versus neutral passengers that simply accumulate through the process of directed evolution. Nevertheless, some mutations do appear to contribute disproportionately to these global changes, including the 381 L insertion and substitutions within the thumb subdomain (G602D and

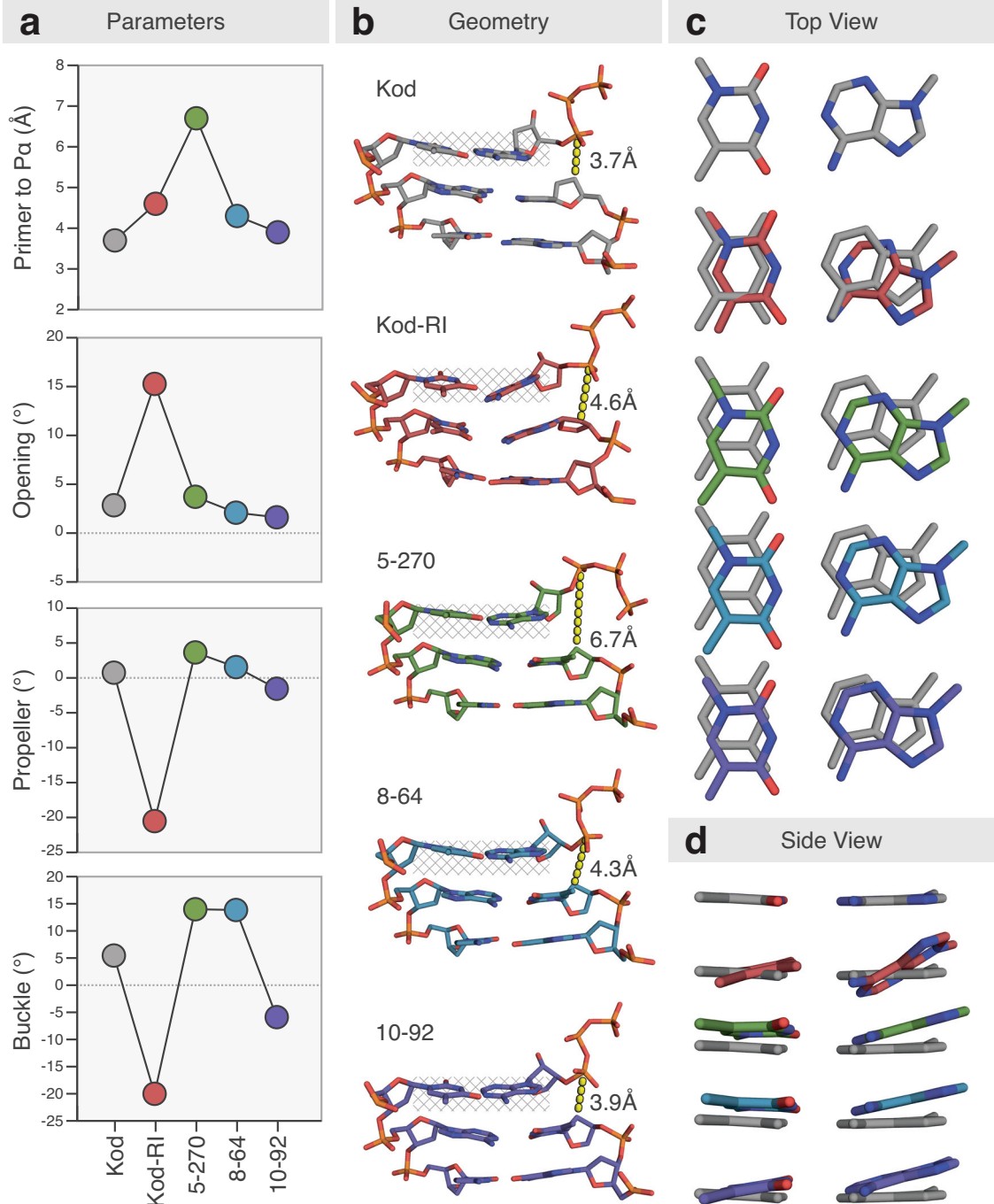

**Fig. 4 | Structural optimization of the nascent base pair and reaction center.**
**a** Nucleic acid structural parameters plotted across generations of TNAP. Measurements reflect changes in the distance from the primer to the a-phosphate and key conformational rotations observed for the nascent base pair formed between the templating base (dT) and incoming triphosphate (dATP or tATP). **b** Structural comparisons of the nascent base pair and reaction center observed for Kod DNAP and generations of TNAPs. DNA and TNA are shown as stick models with the distance from the primer to the a-phosphate denoted with a dashed yellow line and numerical value. Top (**c**) and side (**d**) views of the geometry of the nascent base pair across generations of TNAPs via global structural alignment to Kod DNAP. TNAP generations are color-matched to Fig. 1.

K672R). Future studies aimed at disentangling the functional role of these and other mutations will be critical for elucidating the mechanistic basis of the fidelity improvement observed early in the evolutionary path.

Interpreting the mutations responsible for improved catalytic activity is aided by a single-point reversion analysis performed on the ten amino acid mutations observed between 5–270 and 10–92[11]. This analysis revealed that the top four most important mutations in terms of their contribution to catalytic improvement (D615G, A741P, T548P,

and S493G) involve the gain of either a glycine or proline residue, while the fifth most important mutation (P550H) transitioned a proline to histidine (Figs. 5 and 6). Together, these changes highlight the importance of flexibility and rigidity as a mechanism for fine-tuning the activity of the catalytic center. While these mutations are located -20 Å (except S493G, which is -13 Å) from the active site, defined as the 2′ terminus of the primer, they have a profound impact on enzyme function. The D615G and A741P mutations, first observed in 7–47 and 8–64, respectively, directly contact the primer and template strands,

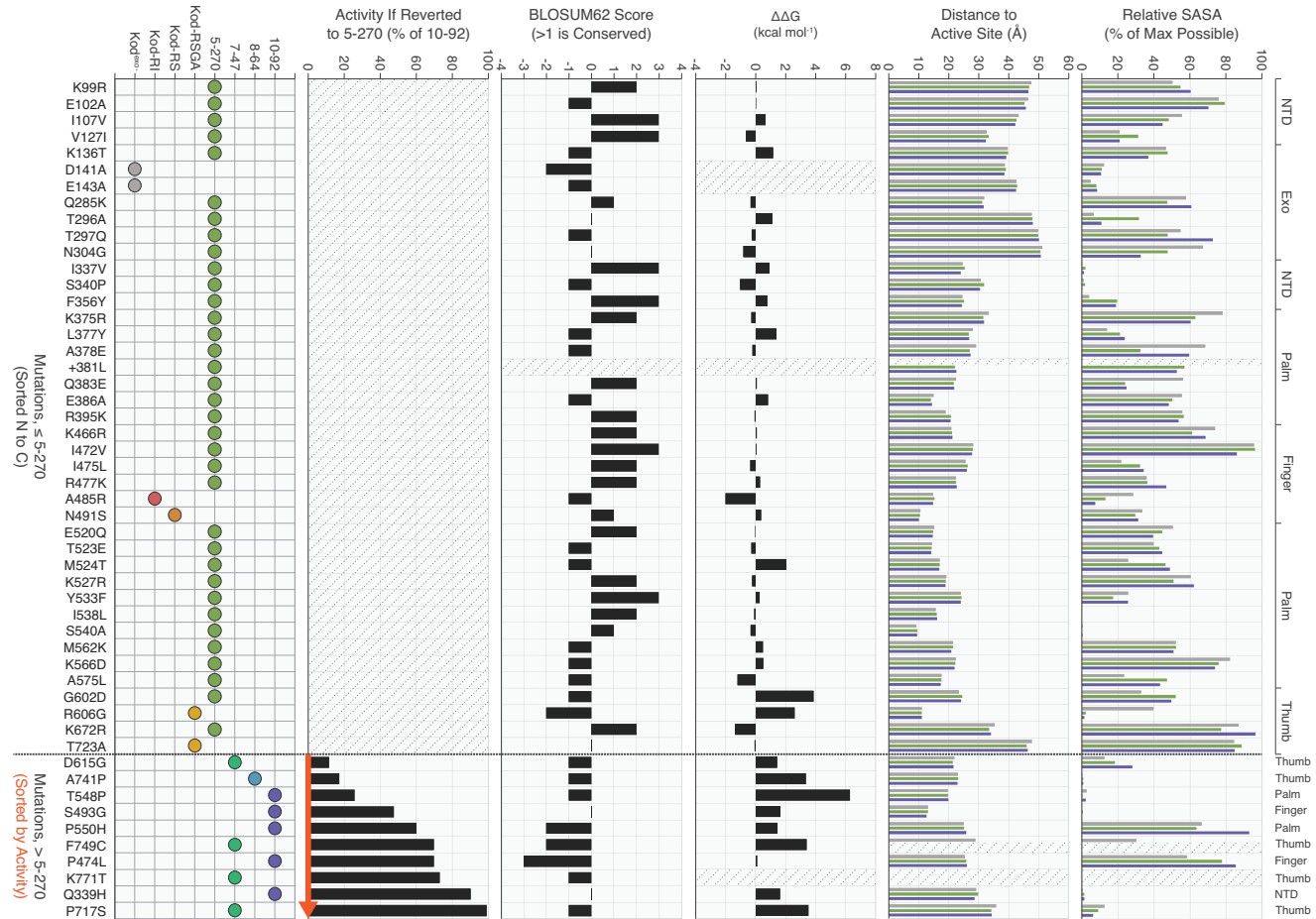

**Fig. 5 | Integrated analysis of 51 mutations acquired during the evolution of 10–92.** Mutations are ordered along the x-axis, with those observed in 5–270 arranged N → C and those acquired after 5–270 separated by a dashed line and ranked by functional importance. For each site, we report the generation of appearance, the effect of reversion on activity (% of 10–92, extracted from prior data[11]), BLOSUM62 score, predicted stability change (ΔΔG, kcal mol⁻¹), distance to the active site, and relative solvent accessibility in ternary complexes (Kod^exo-, gray; 5–270, green; 10–92, purple). Boxes without data or where metrics are not applicable are marked with a gray diagonal dotted pattern.

respectively (Fig. 6; Supplementary Figs. 28 and 29). These interactions balance the need for the enzyme to recognize the duplex tight enough to allow for catalysis yet weak enough to facilitate efficient transloca-tion. The T548P and P550H mutations, both observed in 10–92, induce structural changes in a loop that optimizes the position of the catalytic triad for efficient coordination to divalent metals (Fig. 6 and Supple-mentary Figs. 28–30), allowing for enhanced catalytic efficiency. Finally, the S493G mutation observed in 10–92 is located in the active site pocket (Fig. 6; Supplementary Figs. 28 and 29). This mutation appears to reduce suboptimal interactions between the enzyme and templating base, resulting in better substrate binding and catalysis.

## Discussion

This study provides a high-resolution structural and biochemical view of how directed evolution shapes the structure and function of an enzyme to enable the synthesis of an artificial TNA polymer. By map-ping the evolutionary trajectory of a highly efficient TNAP isolated from a molecular evolution study involving recombination and direc-ted evolution[11], we uncover a series of coordinated molecular events that separate the emergence of fidelity from catalytic efficiency, two properties traditionally thought to be coupled activities in enzyme evolution[15–17].

Contrary to prevailing assumptions, we observe that fidelity and catalysis diverge early in the evolutionary process and are refined through distinct structural pathways. Fidelity improves rapidly

across early variants, corresponding to restoration of Watson–Crick base pair geometry and nascent base pair stability. In contrast, gains in catalytic activity and altered substrate specificity appear only in later generations and correlate with progressive remodeling of the active site architecture and reaction center geometry. These findings suggest that ground-state and transition-state discrimination can evolve independently, offering a modular mechanism by which enzymes acquire new function. A possible reason for this distinction could be that transition-state optimization is more difficult to realize than ground-state optimization, as the free energy of transition-state complexes are much higher than those of ground-state complexes[16].

A defining feature of the evolved polymerase lineage is the adoption of a preorganized, closed finger conformation in the absence of a bound nucleotide. This catalytically competent architecture, which first appears in intermediate 5–270, the most active variant identified after recombination and five rounds of positive selection for TNA synthesis activity, is stabilized by a network of interactions that shift the conformational equilibrium away from the open conforma-tion commonly associated with the binary complex of replicative DNAP. This structural transition may represent a critical inflection point in the evolutionary process, enabling the enzyme to escape the kinetic penalties associated with accommodating the unnatural TNA substrate. Notably, this conformational switch was not predicted by deep learning-based structure prediction models (AlphaFold3),

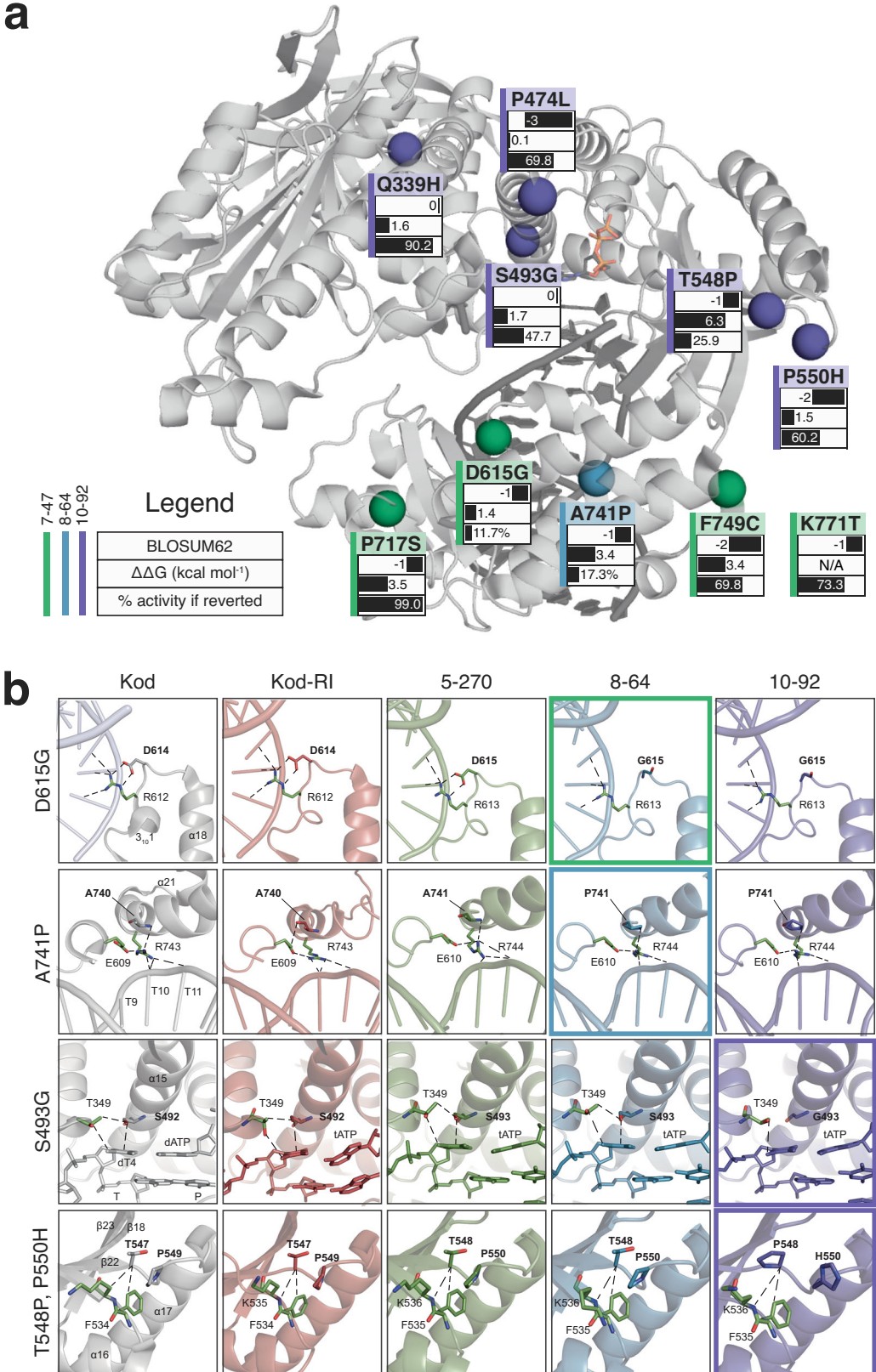

underscoring current limitations in modeling dynamic, evolution-driven enzyme conformations.

Thermal profiling further reveals that even extensive mutational remodeling, 51 mutations in the case of 10–92, does not compromise the structural integrity of the enzyme fold. Instead, the evolved polymerase retains full activity following prolonged exposure at 90 °C. We

attribute this remarkable thermostability to the stringent selection environment imposed by the DrOPS screening workflow, which includes a high-temperature lysis step. These findings challenge the view that functional gains necessarily incur stability trade-offs and highlight the value of integrating environmental constraints into the selection process. Additionally, this observation supports the view that

**Fig. 6 | Structural analysis of key mutations driving improved catalysis. a** Ten mutations acquired during error-prone mutagenesis between 5–270 and 10–92 are mapped onto the Kod$^{exo-}$ scaffold (PDB ID 5OMF) by showing the Cα atoms of each mutated residue as spheres. The K771T mutation near the C-terminus was not resolved in the crystal structure. Labels indicate the mutation, its BLOSUM62 score, predicted ΔΔG (kcal mol$^{-1}$), and the remaining percentage of activity when the mutation is individually reverted. Color coding reflects the generation in which the mutation first appeared (mint green, 7–47; blue, 8–64; purple,10–92). **b** Panels depict key structural changes throughout the evolutionary trajectory by tracking residues impacting activity across crystallized closed ternary structures. Dashed lines indicate inter- and intramolecular interactions with other residues in the protein (green) and the primer-template duplex. Boxes with thicker borders denote the first appearance of the mutations.

intrinsic protein folding stability is an important criterion for promoting protein evolvability[24].

The evolution of substrate preference provides deeper insight into the adaptive landscape navigated by the polymerase. Across the evolutionary trajectory, successive variants exhibited progressively higher TNA synthesis rates, while DNA synthesis activity remained predominant. A marked inversion in substrate specificity emerged only in variant 10–92 (Supplementary Fig. 7). The preference of 10–92 for TNA likely stems from its remodeled active site, whose reduced volume reflects structural adaptation to the smaller size of the TNA triphosphate. This inversion in specificity represents a clear case of functional divergence and illustrates how exploration of sequence space (the ensemble of all possible protein sequences of a given length) can yield enzymes that are not only structurally robust but also precisely tuned for distinct chemical tasks.

Taken together, our findings illustrate one example of how directed evolution can orchestrate long-range structural rearrangements and local active site refinement to generate an enzyme with tailor-made activity. The modular separation of fidelity and catalysis observed here offers a framework for understanding enzyme adaptation and informs future efforts to design polymerases for applications in synthetic biology and medicine. More broadly, this work underscores the importance of combining structural biology with evolutionary trajectories to illuminate mechanisms of molecular innovation that remain opaque to current predictive models.

## Methods
### Reagents
DNA oligonucleotides were purchased from Integrated DNA Technologies (IDT, Coralville, Iowa). TNA triphosphates were obtained by chemical synthesis previously[34,37]. ThermoPol buffer, Q5 DNA polymerase, Xl1 Blue competent *E. coli*, BL21(DE3) competent *E. coli*, Taq DNA polymerase, *NdeI* and *NotI* restriction enzymes were purchased from New England Biolabs (Ipswich, MA). Qiaprep plasmid minikit was purchased from Qiagen (San Diego, CA). Amicon centrifugal filter, hen egg white lysozyme, and chemical reagents including dNTPs, sodium chloride, manganese chloride, magnesium chloride, Tris-HCl, isopropyl β-D-thiogalactoside (IPTG), 1,4-dithiothreitol (DTT), ampicillin, and ammonium persulfate (APS) were purchased from Sigma Aldrich (St. Louis, Missouri). TOPO TA cloning kit, ethylenediaminetetraacetic acid (EDTA), and Dynabeads M270 were purchased from Thermofisher Scientific (Waltham, Massachusetts). SequalGel UreaGel 29:1 Denaturing Gel System was purchased from National Diagnostics (Atlanta, GA). Tetramethyl-ethylenediamine (TEMED) was purchased from Bio-Rad (Hercules, California). Chromatography columns were purchased from Cytiva (Little Chalfont, United Kingdom). EvaGreen® dye was purchased from Biotium (Fremont, CA). Clear V-bottom 96-well plates were purchased from Greiner (Monroe, NC). Plastic 1.5 mL microcentrifuge tubes were purchased from Sigma-Aldrich (St. Louis, MO). Crystallization screens and individual crystallization reagents were purchased from Hampton Research (Aliso Viejo, CA), NeXtal Biosciences (Holland, OH), and Molecular Dimensions (Holland, OH). The Mosquito crystallization robot was purchased from SPT LabTech (Covina, CA). pET-21b(+) plasmid was purchased from Novagen Technology (Glendale, CA). Whole plasmid and Sanger sequencing were performed using Plasmidsaurus (Los Angeles, CA) and Genewiz, respectively (San Diego, CA).

### TNAP expression and purification for biochemical assays
Plasmids for TNAP expression and purification were generated previously[38]. Briefly, a pGDR11 vector containing the TNAP of interest was transformed into XL1 Blue *E. coli*, grown, induced for expression, and harvested. Cells were resuspended and sonicated in TNAP buffer (10 mM Tris-HCl, pH 8.0, 500 mM NaCl, 10% glycerol). The cell lysate was heat-treated at 70 °C for 1 h, cooled on ice for 1 h, and centrifuged (23,708 × $g$, 4 °C, 20 min). The clarified supernatant was treated with a final concentration of 0.5% (v/v) PEI for 15 min and centrifuged (23,708 × $g$, 4 °C, 20 min) to remove nucleic acids. Ammonium sulfate precipitation was performed (final concentration: 60% (w/v)), centrifuged (23,708 × $g$, 4 °C, 20 min), and the protein pellet was resuspended in equilibration buffer (10 mM Tris-HCl, pH 8.0, 50 mM NaCl, 10% glycerol). The TNAP was purified by hand on a 5 mL heparin affinity column and eluted by stepwise addition of buffers containing: 10 mM Tris-HCl, pH 8.0, 10% glycerol and increasing concentrations of NaCl (50, 100, 250, 500, and 750 mM). Eluted TNAP at 500 mM NaCl was pooled and concentrated to 100 μM for biochemical assays.

### TNAP kinetics
TNAP kinetic measurements were performed by monitoring the reaction progress as the DNA primer-template duplex (PBS8: GTCCCCTTGGGG ATACCACC and EM619: CCCACACCCTCCTATCGCTAAACACACACTTA ATAAAGTTGGTGGTATCCCCAAGGGGAC, Supplementary Table 1) was extended with dNTPs or tNTPs to measure DNA or TNA synthesis, respectively, over time[39]. For each time point, 2 individual reaction replicates (15 μL) from a single master mix poised under single-turnover conditions with equimolar concentrations of TNAP (0.2 μM) and preannealed duplex (0.2 μM each, heated at 90 °C for 5 min, cooled on ice) in 1x ThermoPol buffer [20 mM Tris-HCl, 10 mM (NH$_4$)$_2$S)$_4$, 10 mM KCl, 2 mM MgSO$_4$, 0.1% Triton X-100, (pH 8.8)] were pre-equilibrated for 5 min at 55 °C. The reactions were then initiated by adding a preheated triphosphates mixture (200 μM of each). The reactions were stopped at designated time points by plunging the reaction vessel into powdered dry ice. Once all the time points were collected, the frozen reactions were thawed on a cold plate by adding 15 μL (6 μM) EvaGreen® dye, previously identified as the optimal intercalating dye to monitor the synthesis on a DNA template[39]. Each reaction (25 μL) was transferred to a clear V-bottom 96-well plate, and fluorescence intensity was measured (ex: 487/20 nm, em: 528/20 nm) using a Synergy Neo2 plate reader with the Gen5 v3.14.03 software (BioTek).

For both DNA and TNA synthesis, baseline fluorescence was measured using Kod (exo-), the parent polymerase with negligible TNA synthesis ability, which was used to collect a 0-s time point by freezing the reaction immediately after adding tNTPs. The maximum fluorescence value was obtained from a 15-min reaction with each polymerase. The raw fluorescence data from the plate reader were normalized by subtracting the baseline and dividing by the difference between the maximum (15-min reaction) and minimum (baseline) fluorescence values. The fluorescence data were then converted to nucleotides per polymerase[39] by multiplying the data with the conversion factor F = [Template] * L/[Polymerase], where L is the length of extension in bases, then plotted over time. Rates in nucleotides per minute are extracted as the slope of the linear range of each curve. Data were processed using Microsoft Excel v16.66.

## TNAP fidelity

Fidelity analysis was performed in hydrogel particles[40]. In brief, an aliquot of ~12 million hydrogel particles displaying the acrydite PBS8 primer (GTCCCCTTGGGGATACCACC, Supplementary Table 1) was placed into a 1.5 mL and washed twice with 100 µL 1x ThermoPol buffer. The hydrogels were then resuspended in 100 µL 1x ThermoPol buffer containing 2 µM 4NT.9G fidelity template (Supplementary Table 1) and 1% KF-6012 and heated at 95 °C for 3 min. The reaction mixture was then snap-cooled on ice for 5 min to promote primer-template annealing. To initiate TNA synthesis, 2 µL of tNTPs (5 mM stock) and 20 µL of TNAP (10 µM stock) were added, mixed, and then incubated at 55 °C on a ThermoMixer C for 1 h. After the reaction is complete, hydrogels were washed with 100 µL breaking buffer (10 mM Tris-HCl pH 7.5, 100 mM NaCl, 1 mM EDTA, 1% (v/v) SDS, 1% (v/v) Triton X-100) twice, incubated with 100 µL 100 mM NaOH (37 °C, 3 min) to strip the template, and neutralized with 100 µL breaking buffer. The hydrogels were resuspended in 100 µL 1x ThermoPol buffer containing 2 µM PBS7.PBS9 overhang primer (Supplementary Table 1) with a double nucleotide mismatch (TT-TT), 0.01% KF-6012, 3 mM magnesium sulfate and then heated at 95 °C for 3 min. The reaction mixture was then snap-cooled on ice for 5 min to promote primer-template annealing. To initiate the reverse transcription reaction, 10 µL of dNTPs (5 mM stock) and 20 µL of Bst-LF (10 µM stock) were added, mixed, and then incubated at 50 °C on a ThermoMixer C for 4 h. The cDNA was PCR amplified with Taq DNA polymerase using PBS8 and PBS9: CTTTTAAGAACCGGACGAAC as primers (Supplementary Table 1). PCR material was ligated into a TOPO vector using a TOPO-TA kit and cloned into DH5α competent *E. coli* cells. Individual clones were picked, grown in liquid LB media, and sequenced. DNA sequences were aligned and analyzed with the template using CLC Main software. Error rates were determined as $\mu_{exp \to obs} = (\#observed/\#expected) \times 1000$. The total error rate was determined by summing the error rate for each substitution.

## TNAP thermal challenge

5 × 1.5 mL Eppendorf tubes, each containing 100 µL Kod or TNAP (10 µM), were placed into an Eppendorf ThermoMixer set at 90 °C with a thermal lid to prevent evaporation. After 6 h, the tubes were spun at 16,3000 × g for 5 min at 4 °C and UV quantified using a NanoDrop instrument. TNA synthesis was assayed with a primer-template duplex (PBS8_short:/5IRD680/GTCCCCTTGG and EM619: CCCACACCCTCCT ATCGCTAAACACACACTTAATAAAGTTGGTGGTATCCCCAAGGGGAC, Supplementary Table 1). Polymerase activity screen was performed with the following final concentrations: 1 µM primer-template duplex, 100 µM tNTPs, 1 µM heat-treated polymerase in a 20 µL reaction. After 30 min of incubation, a 1 µL aliquot of the reaction was quenched by the addition of 39 µL of quenching buffer and visualized by denaturing polyacrylamide gel electrophoresis with fluorescent imaging on a Li-Cor Odyssey CLx Imager, Image Studio Lite v5.2.

## TNAP cloning, expression, and purification for crystallography

Full-length *tnap* genes in pGDR11 were PCR amplified using the TNAP-Fwd (ATCC*CATATG*ATCCTCGACACTGACTAC) and TNAP-Rvs primers (ACGCATG*CGGCCGC*TCAAGTTCCTTTCGGCGTCAG), Supplementary Table 1, containing *NdeI* and *NotI* restriction enzyme sites, respectively. C-terminal truncation constructs for all TNAPs, except for 8–64, were PCR amplified by substituting the TNAP-Rvs primer with TNAP-Rvs_760 (ACGCATG*CGGCCGC*TCACTTCTGGTAGCGCAGGTC, Supplementary Table 1); 8–64-Rvs_760: ACGCATG*CGGCCGC*TCAAGTTCCTTTCGGCGTCAG was used to amplify 8–64, which harbored acquired mutations at the C-terminal. All reverse primers contain the *NotI* restriction enzyme site. Purified PCR product of each gene construct and pET-21b(+) were digested with *NdeI* and *NotI* restriction enzymes and ligated, resulting in pET21–*5–270*, pET21–*5-270_760* pET21–*7–47*, pET21–*7–47_760*, pET21–*8–64*, pET21–*8–64_760*, and

pET21–*10–92*. Post verification by whole plasmid sequencing, all TNAP constructs were expressed and purified[31]. Briefly, BL21(DE3) cells harboring pET21-*tnap* were grown aerobically at 37 °C in LB medium containing 100 µg mL⁻¹ ampicillin. At an OD₆₀₀ of 0.8, expression of a tagless TNAP was induced with 0.8 mM isopropyl β-D-thiogalactoside at 18 °C for 20 h. Cells were harvested by centrifugation for 20 min at 3315 × g at 4 °C and lysed in 40 mL lysis buffer (10 mM Tris-HCl, pH 7/6.5-full-length/truncated TNAP, 100 mM NaCl, 0.1 mM EDTA, 1 mM DTT, 10% glycerol, 5 mg hen egg white lysozyme) by sonication. The cell lysate was centrifuged at 23,708 × g for 30 min, and the clarified supernatant was heat-treated for 20 min at 70 °C and centrifuged again at 23,708 × g for 30 min. The supernatant was loaded onto 5 mL HiTrap Q HP and heparin HP columns assembled in series with the efflux of the Q column loaded in front of the heparin column. After washing with lysis buffer, the Q column was removed, and TNAP was eluted from the heparin column with a high salt buffer (10 mM Tris-HCl, pH 7/6.5-full-length/truncated TNAP, 1 M NaCl, 0.1 mM EDTA, 1 mM DTT, 10% glycerol) using a linear gradient. Eluted fractions containing TNAP were visualized by SDS-PAGE, pooled, and concentrated using a 30 kDa cutoff Amicon centrifugal filter. Further purification was achieved by size exclusion chromatography (Superdex 200 HiLoad 16/600) pre-equilibrated with TNAP buffer (50 mM Tris-HCl, pH 8.5, 200 mM NaCl, 0.1 mM EDTA, 1 mM DTT). Purified TNAP was concentrated to 20 mg mL⁻¹ for crystallization trials.

## TNAP crystallography

The binary DNA template, T1: /5Cy5/AAACGTACGCAGTTCGCG, was HPLC purified sample, while the remaining oligonucleotides, P: CGCGAACTGCG and T2: TATGCACGTACGCAGTTCGCG, were ordered with standard desalting (Supplementary Table 1). The primer-template duplexes, P/T1 and P/T2 for the binary and ternary complexes, respectively, were prepared by combining equal parts of the respective primer and template strands in TNAP buffer and supplemented with 20 mM MgCl₂, heating at 95 °C for 5 min and slow cooling to 10 °C over 10 min.

All constructs were screened against ~900 conditions in a hanging-drop format using a Mosquito crystallization robot (SPT LabTech). Positive crystal hits were further optimized in 24-well hanging drop trays, with each drop consisting of 1 µL of sample mixed with 1 µL of mother liquor over 500 µL mother liquor in every well. The specific crystallization conditions for each TNAP complex are described below. Six diffraction datasets were collected at synchrotron sources (Advanced Light Source, Stanford Synchrotron Radiation Lightsource, and National Synchrotron Light Source II) from single crystals. Unless specified, images were indexed, integrated, and scaled using XDS[41]. Data collection statistics are summarized in Tables S2 and S3. Initial models were determined by molecular replacement using Phaser[42]; for the binary complexes, the Kod-RSGA binary complex (7RSU) was initially used as the search. However, the 10–92 closed ternary complex (8T3X) was subsequently used due to the closed conformation of the binary structures. 8T3x was used as the search model for the ternary complexes. All final models were determined using iterative rounds of manual building through Coot[43] and refinement with phenix[44]. The stereochemistry and geometry of all structures were validated with Molprobity[45] with the final refinement parameters summarized in Tables S2 and S3. Final coordinates and structure factors have been deposited in the Protein Data Bank.

Binary complexes were prepared by incubating full-length TNAPs (6 mg mL⁻¹) with 1.2 molar equivalents of the annealed P/T1 at 37 °C for 30 min. 5 M excess of tTTP and tATP were added and incubated at 37 °C for 30 min separately. The resulting 5–270 binary complex crystallized in 0.1 M calcium acetate, 0.1 M 2-(N-morpholino)ethanesulfonic acid pH 6, and 15% polyethylene glycol 400 while the remaining binary complexes crystallized in 0.06 M magnesium chloride hexahydrate, 0.06 M calcium chloride dihydrate, 0.1 M imidazole,

0.1 M 2-(N-morpholino) ethanesulfonic acid pH 6, 17.5–19% glycerol, and 7.5–9% polyethylene glycol 4000. The images for the 7–47 binary complex were indexed, integrated, and scaled using iMOSFLM.

Initial binary complexes were prepared by incubating truncated TNAPs (6 mg mL⁻¹) with 1.2 M equivalents of the annealed P/T2 duplex at 37 °C for 30 min. 3 M excess of dtTTP was added and incubated at 37 °C for 30 min. The resulting n + 1 binary complex was further incubated with 10 M excess tATP at 37 °C for another 30 min to yield final ternary complexes. 5–270 ternary complex crystals grew in 0.2 M sodium iodide, 20% polyethylene glycol 3350, and 0.1 M calcium chloride dihydrate, and 8–84 ternary complex crystals grew in 0.2 M sodium iodide, 20% polyethylene glycol 3350, and 0.1 M taurine.

## TNAP structural analysis

Molecular visualizations, RMSD, and distance and torsion angle measurements were performed using licensed PyMOL (version 2.5.2; https://www.pymol.org)[46] and open-source PyMOL (version 3.1.0; https://github.com/schrodinger/pymol-open-source). Structural alignments for visualization and analysis in Figs. 3 and 4 were carried out using the "align" function in PyMOL, applied to the entire protein–nucleic acid complex. Polder omit maps were generated in Phenix using phenix.polder and visualized in PyMOL[47]. Local base-pair and base-pair step parameters are computed using Web 3DNA[48]. 2D interaction maps were assembled with data from Mapiya and DNA-proDB. Pocket volumes were calculated using PyVOL.

## Integrated mutational analysis

Sequence alignments used to generate the fitness landscape (Supplementary Fig. 3) were performed using the EMBOSS Needle algorithm[49] on the EMBL–EBI platform[50]. Conservation was quantified with BLOSUM62 substitution scores[51].

Functional effects of reversions to 5–270 were quantified from the 45-s primer extension assay as previously reported[11]. Band intensities were measured by densitometry in ImageJ[52] as the fraction of full-length product (top band) relative to the total pixel density of each lane. Activities were normalized to the performance of 10–92.

Mutational effects on folding stability were estimated using FoldX5[53] with the BuildModel command applied to a repaired Kod^exo-closed ternary structure (PDB ID: 5OMF). Reported ΔΔG values are in kcal mol⁻¹.

Distances between each mutated alpha carbon atom and the active site, defined by the sugar moiety of the dideoxy primer terminus (C2′ in TNA or C3′ in DNA), were calculated in PyMOL on ternary complexes of Kod, 5–270, and 10–92.

Relative solvent-accessible surface area (SASA) was computed on ternary structures using FreeSASA[54] with the default parameters. Values were reported as relative exposure (%) in ternary complexes of Kod, 5–270, and 10–92.

## AlphaFold3 predictions

AlphaFold3[55] structural predictions were performed on the online server provided by Google DeepMind (https://alphafoldserver.com/) on 2024-08-16, before the initial public release of the 10–92 ternary structure (PDB ID: 8T3X) on 2024-08-28. Two AlphaFold3 runs (denoted AF3-1 and AF3-2) were performed for every evaluated structure, producing 5 models per run (models 0–4, denoted by subscripts). The binary structural predictions were obtained by inputting the sequence of residues 1–776 in the presence of a DNA primer (CGCGAACTGC), DNA template (AAACGTACGCAGTTCGCG) and 2 Mg²⁺ ions. Ternary structural predictions were obtained by inputting the sequence of residues 1–760 in the presence of a DNA primer (CGCGAACTGCGT), DNA template (TATGCACGTACGCAGTTCGCG), an ATP, and 2 Mg²⁺ ions.

## 2′-deoxy-α-L-threofuranosyl thymidine-3′-triphosphate (dtTTP) synthesis

**Synthetic procedures.** All moisture sensitive reactions were performed in anhydrous solvents under an argon or nitrogen atmosphere. All commercial reagents, solvents and anhydrous solvents were used without further purification. Reaction progresses were monitored by thin-layer chromatography using glass-backed analytical Silica Plate with UV-active F254 indicator. Flash column chromatography was performed with Silica Flash® P60 silica gel (40–63 μm particle size) for most of the crude reaction mixture. Yields are reported as isolated yields of pure compounds. ¹H, ¹³C, and ³¹P NMR spectra were analysed on 400 MHz NMR spectrometer (Bruker Avance NEO). ¹H values are reported in parts per million relative to Me₄Si or corresponding deuterium solvents used as an internal standard. ¹³C values are reported in parts per million relative to corresponding deuterium solvents used as an internal standards. ³¹P NMR values are reported in parts per million relative to an external standard of 85% H3PO4. Splitting patterns are designated as follows: s, singlet; d, doublet; dd, doublet of doublets; t, triplet; m, multiplet. High-resolution mass spectrometry data were acquired using the electrospray ionization time-of-flight method at the University of California, Irvine Mass Spectrometry Core Facility.

**Chemical synthesis, step 1.** To a mixture containing the activated nucleoside monophosphate **1**[33] (240 mg, 0.6087 mmol, 1 equiv) and 1.2 equiv of 1-(2-(pyrenesulfonyl)ethyl)pyrophosphate (343 mg, 0.7305 mmol) was added, along with 8 equiv of a premade solution of ZnCl₂ (664 mg, 4.8670 mmol) in 3.2 mL anhydrous DMF under a nitrogen atmosphere. The mixture was stirred at room temperature for 5 h, and the reaction progress was monitored by HPLC (MeCN/0.05 M TEAB buffer, from 0% to 70% over 42 min). After consumption of the starting material, the product was precipitated by adding the reaction mixture dropwise to stirred diethyl ether (170 mL). The precipitate was collected by centrifuging at 20,000 × *g* for 5 min at room temperature. The pellet was resuspended in 30 mL 20% H₂O in MeCN containing 2% *N, N*-diisopropylethylamine (DIPEA) and centrifuged at 20,000 × *g* for 5 min, and the supernatant was collected. The above process was repeated two times, and the combined supernatants were evaporated under diminished pressure. The crude was loaded on a silica gel column (packed with 2% H₂O/isopropanol containing 1% DIPEA) by liquid loading (2 mL of 2% H₂O/isopropanol containing 1% DIPEA) with eluents 5%–10% H₂O/isopropanol containing 1% DIPEA and then 5% H₂O in (isopropanol/MeCN 1:1) containing 1% DIPEA. The fractions containing the product were collected and evaporated under diminished pressure at 30–40 °C to afford pure 240 mg (0.2758 mmol) fully protected nucleoside triphosphate **2**. ³¹P NMR (162 MHz, D₂O) δ −11.93, −12.85 (d, *J* = 15.6 Hz), −23.10.

**Chemical synthesis, step 2.** A solution of fully protected nucleoside triphosphate 240 mg (0.2758 mmol) in 30–33% aqueous NH₄OH was stirred for 18 h at 37 °C in a sealed tube. After the reaction, the solvent was evaporated under diminished pressure. The solid was resuspended with MilliQ water (5 mL) and the aqueous solution was washed with DCM (10 mL) and EtOAc (10 mL). The organic portion was discarded, and the aqueous extract was collected and evaporated under diminished pressure. The crude solid was resuspended with 2 mL of RNAse free water, filtrated by 0.22 μm syringe filter and dropwise added to the forty times volume of acetone (80 mL) at room temperature containing NaClO₄ (390 mg, 3.1815 mmol, 15 equiv). The resulting suspension was centrifuged at 20,000 × *g* for 5 min at room temperature. The supernatant was discarded, and the pellet was washed with organic solution (acetone/DCM, 10:1, 2 × 30 mL) to afford the **1a** (85 mg, 0.1577 mmol, 26% two-step yield). ³¹P NMR (162 MHz, D₂O) δ −8.09, −12.42 (d, *J* = 19.5 Hz), −22.52 (t, *J* = 19.5 Hz, 1H).

## Reporting summary

Further information on research design is available in the Nature Portfolio Reporting Summary linked to this article.

## Data availability

The crystallographic data generated in this study have been deposited in the Protein Data Bank under accession codes 9OAT, 9OAU, 9OAV, 9OAW, 9OAX, and 9OAY. The Protein Data Bank furthermore contains additional structures referenced in this work: 4K8Z, 5OMF, 7OMB, 8T3X. The kinetics, fidelity, and thermostability data generated in this study are provided in the Supplementary Information. Source data are provided as Source data files. Source data are provided with this paper.

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

## Acknowledgements

We wish to thank members of the Chaput lab for helpful comments and suggestions, and the staff at the Advanced Light Source (ALS), Stanford Synchrotron Radiation Lightsource (SSRL), and the National Synchrotron Light Source II (NSLS) for technical assistance. This work was supported in part by the National Science Foundation (2433788) Division of Chemistry of Life's Processes (CLP) and Genetic Mechanisms (GM), and the University of California.

## Author contributions

J.C. and N.C. conceived the study and designed the research plan. M.H., V.M., J.L., M.J.H., R.Q., K.N., K.H., J.M., E.B., and B.B. performed the experiments and analyzed the data. J.C. and N.C. wrote the manuscript with input from all authors.

## Competing interests

J.C., V.M., and the University of California-Irvine have filed a patent application (PCT/US24/11595) on the composition and activity of the 10–92 TNA polymerase. No financial competing interests are declared. The remaining authors declare no competing interests.
