## [Peer Review File · Nature Communications]

Directed Evolution of a TNA Polymerase Identifies Independent Paths to Fidelity and Catalysis

Corresponding Author: Professor John Chaput

Version 0:

Reviewer comments:

Reviewer #1

(Remarks to the Author)

Comments for the authors

This manuscript presents a comprehensive structural and biochemical characterization of the evolutionary trajectory of an engineered polymerase capable of efficiently synthesizing TNA, offering valuable insights into how mutagenesis and selection reshape enzyme architecture and function. These findings not only enhance our understanding of polymerase adaptation but also provide a framework for future protein engineering efforts. By resolving crystal structures of key intermediate variants and correlating them with biochemical data, the authors propose that catalytic efficiency and fidelity can evolve through distinct pathways. While the results are promising, additional clarification and discussion are needed to strengthen some of the conclusions, particularly regarding the mechanistic link between structural changes and functional decoupling. I recommend the manuscript for publication only after the following points have been fully addressed.

Major points

1. In lines 86–87, the manuscript states that “crystal structures were solved as binary and closed ternary complexes, and biochemical data were collected for fidelity, catalytic efficiency, thermal stability, and substrate specificity.” However, limited information is provided about changes in substrate specificity during evolution. Please clarify whether substrate specificity correlates with fidelity or catalytic efficiency.
2. The manuscript emphasizes significant conformational changes in the finger subdomain and active site during evolution. However, it remains unclear how the accumulation of mutations induces these structural rearrangements. Additionally, electron density maps for key residues in the active site should be provided alongside Figures 3 and 4 to support the structural interpretations. In particular, in Figure 4b, the increased distance from the primer to the α -phosphate in the 5-270 variant, accompanied by the divergent conformation of tATP, requires further explanation.
3. In lines 104–109 and 281–282, the authors state: “our analysis shows that fidelity increased substantially in earlier generations of the enzyme, whereas strong improvements in catalytic turnover emerge only during the later stages of evolution. Although not widely discussed in the literature, this observation implies that mutations acquired by directed evolution can differentially affect fidelity and catalysis, challenging the conventional view that these traits evolve in parallel as coupled events.” However, Figure 1 shows that in the later stages of evolution both activity and fidelity increase substantially from variant 8-64 to 10-92, which seems inconsistent with complete decoupling. Could the authors comment on this apparent discrepancy? Additionally, beyond reference 16, are there other studies supporting the “conventional view” of coupled evolution?
4. Variant 10-92, which carries five additional mutations compared to 8-64 (Extended Data Figure 2), shows a sharp increase in activity. How do these mutations lead to local or global structural rearrangements that enhance catalytic efficiency? Are these changes reflected in the active site conformations presented in Figures 3 and 4?
5. Structural alignments are presented in Figures 3 and 4. Please specify whether these alignments were performed using the overall structure, individual protein domains, or DNA/TNA components. Since the alignment strategy can substantially influence the interpretation of structural differences, a clear description of the methodology is necessary.
6. The authors state that “the closed binary motif first appears in the 5-270 intermediate and is absent in the binary structure of Kod-RSGA” (lines 184–185). This suggests that the closed binary motif arises during the transition from Kod-RSGA to 5-270. Please elaborate on how these structural changes contribute to the marked improvement in fidelity observed in the early evolutionary stages. A clearer comparison between the structural features of Kod-RSGA and 5-270 would help clarify the mechanistic basis for this enhancement.

Minor points

1. In Figure 1b, the term “sequence space” should be clearly defined in the legend or the main text.
2. In Supplementary Figure S6, please clarify whether the structure corresponds to wild-type KOD, the 10-92 variant, or

another form. If a previously published structure was used, the relevant PDB code should be provided. In addition, the description in lines 150–152 is currently unclear and should be revised for clarity.

3. In Figures 2f, 2g, 2h, and S8, all interactions are depicted using dashed lines. However, some represent hydrogen bonds and others hydrophobic interactions. These should be distinguished visually to avoid confusion. For example, hydrogen bonds could be shown as dashed lines, while hydrophobic interactions should be depicted differently.

4. In Figure 2g, the residue labeled T267 appears to actually be I268. Please verify and correct.

Reviewer #2

(Remarks to the Author)

Hajjar and colleagues present an extensive crystallographic analysis of directed evolution intermediates found in the development of an efficient TNA synthetase. I put below my comments roughly in the order they appear in the manuscript, with more important comments highlighted (*).

*1. Difference between $n=1$ and generalizations. The authors repeatedly claim they are demonstrating evolutionary principles based on the single evolutionary path they analysed, without discussing how the selection pressure may have influenced the evolutionary path. I don't think that is needed and anchoring the path to polymerase biochemistry will not detract from the novelty of the paper.

*2. The differential impact on fidelity and catalysis for polymerases is widely discussed in the literature, but rarely in B-family thermophilic proteins. There is extensive literature available on Pol beta, which I tried to summarise (and integrate with DE efforts) here: [10.1016/j.copbio.2018.11.008](https://doi.org/10.1016/j.copbio.2018.11.008).

3. Impact of 51 mutations. While I agree that the screening methodology actively excludes polymerases that are no longer thermostable, I think the nature and location of the mutations can also be taken into consideration. Surface residues (therefore not part of significant contact networks) and conservative mutations that are present in similar thermostable proteins, would not be expected to significantly affect protein stability.

4. TNAP kinetics. It would be useful to add the reaction buffer composition rather than rely on reference to previous work. I also wondered if the authors ever tried to optimise the reaction buffer (beyond supplementing reactions with Mn^{2+}). Some of the enzymes I have worked with had shifted optima Mg concentration, which is often ignored by some in the field.

*5. TNAP Fidelity. The substrate used to determine fidelity has biases (no GG, known to affect fidelity of TNA polymerases), therefore the values obtained do not sample all possible dinucleotide transitions and may be overestimating fidelity. In addition, based on our own calculations, 1000 bases are not enough data to claim differences between 99.1% and 99.9% incorporation accuracy. What are the confidence intervals for the determined fidelities?

*6. Crystallization poses. While I find very interesting that the evolutionary intermediates crystallized in a "closed" conformation, can this not simply be a crystallization artifact? How do the crystallization conditions of these variants compare to the conditions the starting variant (Kod-RI/Kod-RSGA) was crystallized? Have the authors ever explored how these proteins behave in solution? For the avoidance of doubt, I think the authors need to acknowledge and discuss this point in the manuscript. I am not asking for single-molecule FRET experiments which would be the only way to demonstrate that.

*7. Link between fidelity and catalysis. Given the title, I looked forward to a more extended discussion on these differences for polymerases. If the crystallization pose is relevant in solution, pre-organising binary complex could suggest that the initial gains in TNA incorporation are coming from stabilising the closed conformation of the enzyme. That would be expected to reduce the fidelity of the polymerase, unless that step is also affecting the enzyme's processivity (measured as bound time rather than number of incorporations per binding). In line with that comment, I disagree that the prevailing assumption in the field is that fidelity and catalysis go hand-in-hand for polymerases.

8. Figure 3 (but also applicable to other single amino acid superposition figures). Presenting the relative positioning of the residues without making reference to the electron density and B-factor of the residue in those positions in the crystal reduces the quality of the comparison. Including those directly in the figure may make it too busy, but that could be included in SI.

*9 "...importance of long-range structural... geometry". Unfortunately, ref. 29 does not support that statement. Ref. 29 simply lists how residues are in non-sequential networks and the size and distance may depend on the local structure.

10. AlphaFold. It may be worth including in the main text that AlphaFold3 was used and tempering the statement. AlphaFold2 was trained differently and would have been far more susceptible to homology matches – justifying the side chain positioning as per training data. AlphaFold3 should be less susceptible to homology matches but it provides a static pose, much like crystallography. Relaxing the poses can have a significant impact on the argument that AI cannot deliver accurate structures (or at least meaningful structural insights).

Vitor Pinheiro

Reviewer #3

(Remarks to the Author)

Version 1:

Reviewer comments:

Reviewer #1

(Remarks to the Author)

The authors have thoroughly addressed most of the concerns raised in the initial review, and the manuscript has been significantly strengthened. The study presents valuable insights, and I recommend acceptance after final revision. The following are two remaining minor suggestions to further improve the manuscript:

1. In Figure 2, short dashed lines denote electrostatic interactions, while long dashed lines show hydrophobic interactions, which helps distinguish between different types of interactions. This notation should be applied consistently in Figure 6 and Supplementary Figure S12.

2. In the response, the authors state that "substrate specificity refers to the ability of the polymerase to distinguish TNA versus DNA substrates based on their catalytic rates of incorporation." Supplementary Figure S7 shows a sharp rise in TNA catalytic activity between variants 8-64 and 10-92, which coincides with a notable enhancement in substrate specificity. To elucidate the structural basis for the evolutionary changes in fidelity and catalytic activity, the manuscript offers a thorough structural analysis of the catalytic core, including active site organization, interactions and so on, while lack of structural comparison regarding how the enzyme's interactions with dA and tA change during the evolution from the dA to tA substrate. Including such an analysis is essential to fully clarify the mechanism.

Reviewer #2

(Remarks to the Author)

I thank the authors for the changes introduced in the manuscript.
Points raised here (where possible) use the same numbering as in the original comments.

#5 - TNAP fidelity: This point remains unaddressed. The field's common practice has shifted significantly in the last decade. If the authors persist in applying this approach, it would still benefit from a statistical test to measure the significance of the findings. A test to compare 2 Poisson distributions (using individual reads as the estimate of an error rate) would be the bare minimum.

#7 - fidelity vs. catalysis. Based on my own experience with Tgo, KOD and Phi29 evolution campaigns, gains in processivity dominate early detection in selection campaigns. Here, increases in fidelity delivered processivity faster than improved catalysis as it would be expected from the search pattern, which started from homologous recombinations. Nevertheless, this point is academic and requires no change to the manuscript.

#9 - While I agree that the review is of importance, it does not support the statement being made by the authors that "long-range interactions support the fine-tuning of active site geometry". My particular disagreement here is focused on the use of "fine-tuning". The review supports that long-range interactions can be implicated in catalysis and functional specificity.

I don't expect further changes to the manuscript, but I leave it to the editors whether the points raised here would need to be addressed before publication.

Reviewer #3

(Remarks to the Author)

Version 2:

Reviewer comments:

Reviewer #2

(Remarks to the Author)

I thank the authors for adding the statistical test. It would have been good to include that result also in Figure 1d. Other than that, I think the research is ready for publication.
Best wishes,

REVIEWER 1

This manuscript presents a comprehensive structural and biochemical characterization of the evolutionary trajectory of an engineered polymerase capable of efficiently synthesizing TNA, offering valuable insights into how mutagenesis and selection reshape enzyme architecture and function. These findings not only enhance our understanding of polymerase adaptation but also provide a framework for future protein engineering efforts. By resolving crystal structures of key intermediate variants and correlating them with biochemical data, the authors propose that catalytic efficiency and fidelity can evolve through distinct pathways. While the results are promising, additional clarification and discussion are needed to strengthen some of the conclusions, particularly regarding the mechanistic link between structural changes and functional decoupling. I recommend the manuscript for publication only after the following points have been fully addressed.

Response: We thank the reviewer for their positive assessment of the study.

Major points

1. In lines 86–87, the manuscript states that “crystal structures were solved as binary and closed ternary complexes, and biochemical data were collected for fidelity, catalytic efficiency, thermal stability, and substrate specificity.” However, limited information is provided about changes in substrate specificity during evolution. Please clarify whether substrate specificity correlates with fidelity or catalytic efficiency.

Response: In the context of this study, substrate specificity refers to the ability of the polymerase to distinguish TNA versus DNA substrates based on their catalytic rates of incorporation. The main text was updated to clarify this point and Supplementary Figure 7 was revised with an additional panel showing the rates of synthesis along with a matching bar graph of substrate specificity.

2. The manuscript emphasizes significant conformational changes in the finger subdomain and active site during evolution. However, it remains unclear how the accumulation of mutations induces these structural rearrangements. Additionally, electron density maps for key residues in the active site should be provided alongside Figures 3 and 4 to support the structural interpretations. In particular, in Figure 4b, the increased distance from the primer to the α -phosphate in the 5-270 variant, accompanied by the divergent conformation of tATP, requires further explanation.

Response: Supplementary Figures 1 and 2 were added as new images showing global changes in the protein architecture between 10-92 and its closest natural homolog, Kod DNA polymerase. The structural differences first appeared in 5-270 and are retained in 10-92. This pattern suggests that the changes arose through recombination and represent a defining structural signature of the engineered polymerase.

Additionally, the manuscript was revised to include a discussion of the mutational landscape underlying the activity of 10-92. This new section, and accompanying figures, describe the mutations and/or structural changes responsible for improved fidelity and catalytic activity.

Last, the addition of polder maps to Figures 3 and 4 yielded a change to the image that was too small to visualize at 100% magnification. Recognizing the importance of this information, we prepared supplementary figures showing the electron density of key residues in the active site, as well as electron density of the nascent base pair (dT:tATP) and adjacent template-primer region. In the case of 5-270, we analyzed a second crystal structure with an alternate tATP conformation, which provided a primer to α -phosphate distance of 5.7 Å. The main text was updated with this new information, which we attribute to flexibility within the increased size of the active site pocket.

3. In lines 104–109 and 281–282, the authors state: “our analysis shows that fidelity increased substantially in earlier generations of the enzyme, whereas strong improvements in catalytic turnover emerge only during the later stages of evolution. Although not widely discussed in the literature, this observation implies that mutations acquired by directed evolution can differentially affect fidelity and catalysis, challenging the conventional view that these traits evolve in parallel as coupled events.” However, Figure 1 shows that in the later stages of evolution both activity and fidelity increase substantially from variant 8-64 to 10-92, which seems inconsistent with complete decoupling. Could the authors comment on this apparent discrepancy? Additionally, beyond reference 16, are there other studies supporting the “conventional view” of coupled evolution?

Response: We did not intend to suggest that fidelity and catalysis are fully decoupled, but rather that substantial improvements in each trait arise at different stages of the directed evolution pathway. As the reviewer correctly notes, the strong gain in catalytic activity coincides with a second increase in fidelity. To clarify this point, we have re-written this section of the results to temper our language and better frame the relationship between fidelity and catalysis. Finally, additional references were added to support the claim that fidelity is expected to improve with faster catalysis.

4. Variant 10-92, which carries five additional mutations compared to 8-64 (Extended Data Figure 2), shows a sharp increase in activity. How do these mutations lead to local or global structural rearrangements that enhance catalytic efficiency? Are these changes reflected in the active site conformations presented in Figures 3 and 4?

Response: A single-point reversion analysis described previously implicates the T548P mutation and to a lesser extent the P550H mutation as sequence changes that are likely responsible for improved catalytic activity. Since global architectural changes were largely fixed at 5-270, these mutations appear to fine tune the active site through their proximity to the catalytic aspartates (D405, D541, and D543). The manuscript and supplementary material were updated to clarify this point.

5. Structural alignments are presented in Figures 3 and 4. Please specify whether these alignments were performed using the overall structure, individual protein domains, or DNA/TNA components. Since the alignment strategy can substantially influence the interpretation of structural differences, a clear description of the methodology is necessary.

Response: All structural alignments were performed by global alignment of the C-alpha residues. The figure legends and Methods were updated to clarify this point.

6. The authors state that “the closed binary motif first appears in the 5-270 intermediate and is absent in the binary structure of Kod-RSGA” (lines 184–185). This suggests that the closed binary motif arises during the transition from Kod-RSGA to 5-270. Please elaborate on how these structural changes contribute to the marked improvement in fidelity observed in the early evolutionary stages. A clearer comparison between the structural features of Kod-RSGA and 5-270 would help clarify the mechanistic basis for this enhancement.

Response: The manuscript was revised to include a new supplementary figure (Supplementary Figure 12), comparing the structures of Kod-RSGA and 5-270. Additionally, the results section was modified to include a discussion of the mutational landscape underlying the activity of 10-92, which delineates mutational and/or structural changes impacting fidelity and activity.

Minor points

1. In Figure 1b, the term “sequence space” should be clearly defined in the legend or the main text.

Response: The text was revised to define the term ‘sequence space’.

2. In Supplementary Figure S6, please clarify whether the structure corresponds to wild-type KOD, the 10-92 variant, or another form. If a previously published structure was used, the relevant PDB code should be provided. In addition, the description in lines 150–152 is currently unclear and should be revised for clarity.

Response: The legend for Supplementary Figure 8 was revised to clarify that the structure shown corresponds to that of wildtype Kod DNA polymerase (PDB ID: 5OMF).

3. In Figures 2f, 2g, 2h, and S8, all interactions are depicted using dashed lines. However, some represent hydrogen bonds and others hydrophobic interactions. These should be distinguished visually to avoid confusion. For example, hydrogen bonds could be shown as dashed lines, while hydrophobic interactions should be depicted differently.

Response: Figure 2 (panels f, g, and h) and Supplementary Figure 12 were updated to delineate the different types of interactions observed in the active site.

4. In Figure 2g, the residue labeled T267 appears to actually be I268. Please verify and correct.

Response: The figure was updated to correct the mistake.

REVIEWER 2

Hajjar and colleagues present an extensive crystallographic analysis of directed evolution intermediates found in the development of an efficient TNA synthetase. I put below my comments roughly in the order they appear in the manuscript, with more important comments highlighted (*).

Response: We thank the reviewer for their positive assessment of the study.

*1. Difference between $n=1$ and generalizations. The authors repeatedly claim they are demonstrating evolutionary principles based on the single evolutionary path they analysed, without discussing how the selection pressure may have influenced the evolutionary path. I don't think that is needed and anchoring the path to polymerase biochemistry will not detract from the novelty of the paper.

Response: We agree that our analysis is based on a single evolutionary trajectory, and we were careful to acknowledge this limitation in the revised manuscript. Our goal was to use this trajectory as an example of how evolutionary pressures can shape the emergence of specific biochemical traits in a polymerase scaffold. The manuscript was revised to clarify this point by discussing the imposed selection pressure, which in this case, involved progressively reducing the incubation time to favor improved catalysis. We agree that future studies analyzing alternate trajectories will be important to assess the generality of our observations.

*2. The differential impact on fidelity and catalysis for polymerases is widely discussed in the literature, but rarely in B-family thermophilic proteins. There is extensive literature available on Pol beta, which I tried to summarise (and integrate with DE efforts) here: 10.1016/j.copbio.2018.11.008.

Response: The manuscript was revised to include a discussion of the differential impact of mutations on polymerase fidelity and catalytic activity. This included a new section on the mutational landscape

underlying the activity of 10-92 as well as accompanying figures, describing the mutations and/or structural changes responsible for improved fidelity and catalytic activity.

3. Impact of 51 mutations. While I agree that the screening methodology actively excludes polymerases that are no longer thermostable, I think the nature and location of the mutations can also be taken into consideration. Surface residues (therefore not part of significant contact networks) and conservative mutations that are present in similar thermostable proteins, would not be expected to significantly affect protein stability.

Response: The manuscript was revised to include a deeper analysis of the mutations responsible for improved fidelity versus those responsible for improved catalysis. In addition to textual changes, this includes a new main text figure summarizing an integrated analysis of each mutation.

4. TNAP kinetics. It would be useful to add the reaction buffer composition rather than rely on reference to previous work. I also wondered if the authors ever tried to optimise the reaction buffer (beyond supplementing reactions with Mn^{2+}). Some of the enzymes I have worked with had shifted optima Mg concentration, which is often ignored by some in the field.

Response: The methods section was revised to include the composition of ThermoPol buffer from NEB, which was used for all polymerase reaction described in the study. We have not attempted to optimize the reaction buffer further, but could consider this option in the future.

*5. TNAP Fidelity. The substrate used to determine fidelity has biases (no GG, known to affect fidelity of TNA polymerases), therefore the values obtained do not sample all possible dinucleotide transitions and may be overestimating fidelity. In addition, based on our own calculations, 1000 bases are not enough data to claim differences between 99.1% and 99.9% incorporation accuracy. What are the confidence intervals for the determined fidelities?

Response: The template used for the fidelity assay was chosen to allow comparison with our past work. The sequencing results indicate a clear improvement in fidelity as the polymerase progresses along the evolutionary trajectory. While future studies may explore this question in more detail, for example, using NGS approaches to evaluate the fidelity of TNA replication from thousands of different templates, as was the case for our earlier study on information storage, we believe that the current experiment adequately addresses the fidelity of 10-92. We also wish to point out that it is common practice in the field to perform this assay on a single template with data deriving from 1000 nucleotide incorporation events.

*6. Crystallization poses. While I find very interesting that the evolutionary intermediates crystallized in a “closed” conformation, can this not simply be a crystallization artifact? How do the crystallization conditions of these variants compare to the conditions the starting variant (Kod-RI/Kod-RSGA) was crystallized? Have the authors ever explored how these proteins behave in solution? For the avoidance of doubt, I think the authors need to acknowledge and discuss this point in the manuscript. I am not asking for single-molecule FRET experiments which would be the only way to demonstrate that.

Response: We agree that crystallographic artifacts should always be considered when evaluating new protein conformations by X-ray crystallography. However, in this case, there are multiple lines of evidence arguing against the closed binary conformation being an artifact of crystallization. First, binary structures of the TNAP variants in question (5-270, 7-47, 8-64, and 10-92) crystallized under two distinct precipitation conditions. The reoccurrence of this conformation under two different crystallization conditions suggests that the conformation reflects a bone fide structural feature rather than a condition-specific artifact. Second, the closed binary conformation is absent in Kod-RI and

Kod-RSGA, further strengthening the view that the conformational difference arises from sequence changes rather than crystallization chemistry. Finally, the closed binary conformation emerges in variants that also exhibit changes in their biochemical activity, providing a functional correlation that supports its structural relevance. The manuscript was updated to clarify this point. No solution-based experiments have been undertaken to further explore this point, but could be worthwhile exercises in the future.

*7. Link between fidelity and catalysis. Given the title, I looked forward to a more extended discussion on these differences for polymerases. If the crystallization pose is relevant in solution, pre-organising binary complex could suggest that the initial gains in TNA incorporation are coming from stabilising the closed conformation of the enzyme. That would be expected to reduce the fidelity of the polymerase, unless that step is also affecting the enzyme's processivity (measured as bound time rather than number of incorporations per binding). In line with that comment, I disagree that the prevailing assumption in the field is that fidelity and catalysis go hand-in-hand for polymerases.

Response: The manuscript was revised to clarify the relationship between fidelity and catalysis. The prevailing view is that a selection for improved catalytic activity through the use of progressively shorter incubation times should, indirectly, enrich for enzymes with higher fidelity. This is because less faithful polymerases are more likely to stall at misincorporation sites, thereby failing to complete the extension within the allotted time. Thus, variants that maintain rapid synthesis under stringent incubation times are expected to combine enhanced catalytic efficiency with increased fidelity.

8. Figure 3 (but also applicable to other single amino acid superposition figures). Presenting the relative positioning of the residues without making reference to the electron density and B-factor of the residue in those positions in the crystal reduces the quality of the comparison. Including those directly in the figure may make it too busy, but that could be included in SI.

Response: The manuscript was updated to include polder maps of key residues and nucleotide conformations in the active site. These data provide confidence that the observed structural changes correspond to regions of high experimental confidence.

*9 "...importance of long-range structural... geometry". Unfortunately, ref. 29 does not support that statement. Ref. 29 simply lists how residues are in non-sequential networks and the size and distance may depend on the local structure.

Response: Reference 29 (now ref 35 in the revised manuscript) is a review on the importance of long-range interactions in protein folding.

10. AlphaFold. It may be worth including in the main text that AlphaFold3 was used and tempering the statement. AlphaFold2 was trained differently and would have been far more susceptible to homology matches – justifying the side chain positioning as per training data. AlphaFold3 should be less susceptible to homology matches but it provides a static pose, much like crystallography. Relaxing the poses can have a significant impact on the argument that AI cannot deliver accurate structures (or at least meaningful structural insights).

Response: The Results and Methods sections of the manuscript were revised to clarify that Alpha Fold3 was used for the analysis. Additionally, we redesigned Supplementary Figure 15 and added Supplementary Figure 17 to visualize AlphaFold3 predictions.

Vitor Pinheiro

REVIEWER 3

Response: We thank the reviewer for their positive assessment of the study. Their comments were addressed in one of the reports provided above.

RESPONSE TO THE REVIEWER'S COMMENTS

Reviewer #1:

The authors have thoroughly addressed most of the concerns raised in the initial review, and the manuscript has been significantly strengthened. The study presents valuable insights, and I recommend acceptance after final revision. The following are two remaining minor suggestions to further improve the manuscript:

1. In Figure 2, short dashed lines denote electrostatic interactions, while long dashed lines show hydrophobic interactions, which helps distinguish between different types of interactions. This notation should be applied consistently in Figure 6 and Supplementary Figure S12.

Response: Figure S12 was updated to include long dashes for the hydrophobic interactions. However, we decided not to change Figure 6, because some of the lines, especially those between amino acid side chains and DNA bases, contain both hydrophobic and electrostatic interactions.

2. In the response, the authors state that "substrate specificity refers to the ability of the polymerase to distinguish TNA versus DNA substrates based on their catalytic rates of incorporation." Supplementary Figure S7 shows a sharp rise in TNA catalytic activity between variants 8-64 and 10-92, which coincides with a notable enhancement in substrate specificity. To elucidate the structural basis for the evolutionary changes in fidelity and catalytic activity, the manuscript offers a thorough structural analysis of the catalytic core, including active site organization, interactions and so on, while lack of structural comparison regarding how the enzyme's interactions with dA and tA change during the evolution from the dA to tA substrate. Including such an analysis is essential to fully clarify the mechanism.

Response: We agree that it would be interesting to study how the enzyme changed its substrate specificity from DNA to TNA. However, such an analysis would require crystal structures of the evolved enzymes with the natural substrate, which is beyond the scope of the current study. Nevertheless, we do hope to provide such information in a future study.

Reviewer #2:

I thank the authors for the changes introduced in the manuscript.
Points raised here (where possible) use the same numbering as in the original comments.

#5 - TNAP fidelity: This point remains unaddressed. The field's common practice has shifted significantly in the last decade. If the authors persist in applying this approach, it would still benefit from a statistical test to measure the significance of the findings. A test to compare 2 Poisson distributions (using individual reads as the estimate of an error rate) would be the bare minimum.

Response: We updated Figure S6 to include statistical data with and without insertions and deletions (indel). The comparison between Kod-RSGA and 5-270 with and without indels was significant in both cases. This result is consistent with the interpretation that fidelity improved during the early stages of evolution, while gains in catalysis arose later in the selection. For consistency, the axis in Figure 1d was also updated to include the fidelity value with indels.

#7 - fidelity vs. catalysis. Based on my own experience with Tgo, KOD and Phi29 evolution campaigns, gains in processivity dominate early detection in selection campaigns. Here, increases in fidelity delivered processivity faster than improved catalysis as it would be expected from the search pattern, which started from homologous recombinations. Nevertheless, this point is academic and requires no change to the manuscript.

Response: We are not aware of any prior work in the field where kinetic rates and fidelity have been examined in detail across an evolutionary trajectory for an engineered polymerase, and as such, have no basis of comparison for a competing model.

#9 - While I agree that the review is of importance, it does not support the statement being made by the authors that "long-range interactions support the fine-tuning of active site geometry". My particular disagreement here is focused on the use of "fine-tuning". The review supports that long-range interactions can be implicated in catalysis and functional specificity.

Response: We do not feel that this is a valid concern, as the mutations are nearly all located at long-distance positions and their accumulation in the protein sequence resulted in changes in the enzyme active site that were visualized by protein X-ray crystallography. Thus, our use of the term "fine-tuning" is consistent with other examples in the literature, where mutations accumulated by directed evolution leads to improvements in substrate recognition.

I don't expect further changes to the manuscript, but I leave it to the editors whether the points raised here would need to be addressed before publication.

Reviewer #3:

Response: We thank the reviewer for their positive assessment of the manuscript.